# Emergence of linkage between cooperative RNA replicators encoding replication and metabolic enzymes through experimental evolution

**Kensuke Ueda[1], Ryo Mizuuchi [2,3]\*, Norikazu Ichihashi[1,4,5]\***

**1** Department of Life Science, Graduate School of Arts and Science, the University of Tokyo, Meguro, Tokyo, Japan, **2** Department of Electrical Engineering and Bioscience, Faculty of Science and Engineering, Waseda University, Shinjuku, Tokyo, Japan, **3** JST, FOREST, Kawaguchi, Saitama, Japan, **4** Komaba Institute for Science, the University of Tokyo, Meguro, Tokyo, Japan, **5** Universal Biology Institute, the University of Tokyo, Meguro, Tokyo, Japan

\* mizuuchi@waseda.jp (RM); ichihashi@bio.c.u-tokyo.ac.jp (NI)

**Data Availability Statement:** All relevant data are within the paper and its Supporting Information files.

## Abstract

The integration of individually replicating genes into a primitive chromosome is a key evolutionary transition in the development of life, allowing the simultaneous inheritance of genes. However, how this transition occurred is unclear because the extended size of primitive chromosomes replicate slower than unlinked genes. Theoretical studies have suggested that a primitive chromosome can evolve in the presence of cell-like compartments, as the physical linkage prevents the stochastic loss of essential genes upon division, but experimental support for this is lacking. Here, we demonstrate the evolution of a chromosome-like RNA from two cooperative RNA replicators encoding replication and metabolic enzymes. Through their long-term replication in cell-like compartments, linked RNAs emerged with the two cooperative RNAs connected end-to-end. The linked RNAs had different mutation patterns than the two unlinked RNAs, suggesting that they were maintained as partially distinct lineages in the population. Our results provide experimental evidence supporting the plausibility of the evolution of a primitive chromosome from unlinked gene fragments, an important step in the emergence of complex biological systems.

## Author summary

The integration of genes into a chromosome is a fundamental genetic organization in all extant life. The assembly of unlinked genes during prebiotic evolution was likely a major evolutionary transition toward the development of a complex cell. Decades of theoretical studies have suggested a plausible evolutionary pathway to a primitive chromosome from replicating RNA molecules that harbor cooperative genes within a protocell structure. However, demonstrating the evolution of a primitive chromosome in an experimental setup is challenging. We previously developed a cooperative RNA replication system in which two types of RNAs co-replicate using their self-encoded replication and metabolic

**Funding:** This work was supported by Japan Society for the Promotion of Science KAKENHI (21H05867 and 23H04403 to R.M., 22H05402 to N.I.) and Japan Science and Technology Agency PRESTO (JPMJPR19KA to R.M.) and CREST (JPMJCR20S1 to N.I.). The funders had no role in study design, data collection and analysis, decision to publish, or preparation of the manuscript.

**Competing interests:** The authors have declared that no competing interests exist.

enzymes. Using this system, in the present study, we demonstrate the evolution of a linkage between the two cooperative RNA replicators in compartments. An evolved "linked" RNA harbored the entire region of both genes, accumulated distinct mutations, and retained the ability to replicate using the two proteins translated from itself. These experimental findings support a prebiotic evolutionary scenario, in which unlinked genes assembled into a single genomic structure.

## Introduction

All extant cells have chromosomes that harbor multiple genes and ensure their organized inheritance. In the early evolution of life, such a genome organization may have been absent; unlinked RNA molecules encoding different functions or genes cooperated with each other for replicating the entire system [1–6]. The subsequent appearance of a linkage between cooperative RNA replicators, or a primitive chromosome, is considered a major evolutionary transition toward complex biological organization [2–4]. Chromosome formation could have been both advantageous and disadvantageous for primitive life. Chromosomes enabled the synchronized replication of essential genes and potentially drove the evolution of efficient enzymes [7]. However, chromosomes were longer than unlinked RNA and hence required more time to replicate and had a higher chance of degradation. It remains unclear whether a primitive chromosome could evolve from individual cooperating RNA molecules despite these disadvantages.

Theoretical studies have suggested that the presence of primitive cell-like compartments (protocells) could have facilitated the selection of a primitive chromosome over individual RNA replicators [8–11]. The random assortment of cooperatively replicating RNAs upon protocell division may have caused the stochastic loss of cooperating partners. In contrast, the formation of a primitive chromosome ensured the co-encapsulation of cooperative genes, and therefore, a chromosome could have had an evolutionary advantage over unlinked replicators in compartments [8,9,11]. Another advantage is that chromosome formation eliminates the competition between unlinked replicators for the same resources. Although previous studies have established theoretical frameworks for the evolution of a primitive chromosome, empirical demonstration is lacking.

Previously, we constructed an experimental cooperative RNA replication system consisting of two RNAs and a reconstituted translation system [12]. One of the RNAs encodes a subunit of Qβ replicase, an RNA-dependent RNA polymerase derived from Qβ phage, and the other encodes a metabolic enzyme, nucleotide diphosphate kinase (NDK). We found that the cooperative RNAs can sustainably replicate and evolve in microscale water-in-oil droplets.

In the present study, we examined whether a linked RNA could evolve through the long-term replication of the two cooperative RNAs in water-in-oil droplets. We conducted two successive long-term replication experiments by gradually increasing RNA concentrations, since higher RNA concentrations raise the likelihood of generating linked RNAs through recombination or ligation, which could be facilitated by the RNA polymerase [13–15] or occur spontaneously [16–18]. We found that linked RNAs comprising the two cooperating RNAs appeared during the experiment. Sequence analyses revealed that they harbored complete genes, as well as accumulated mutations that were distinct from those in the cooperative RNA fragments. Subsequent biochemical analysis confirmed that an emergent linked RNA replicated the entire sequence by expressing both encoded proteins. Our study provides experimental evidence that

supports an evolutionary transition scenario of individually replicating genes to a primitive multi-gene chromosome.

## Results

### Cooperative RNA replication system

The cooperative RNA replication system consists of a reconstituted cell-free translation system derived from *Escherichia coli* [19] and two types of cooperative RNA replicators (Rep- and NDK-RNAs) (Fig 1A) [12]. Rep-RNA (2041 bp) encodes the catalytic subunit of Qβ replicase (*rep* gene), which becomes active upon association with EF-Tu and EF-Ts in the translation system. NDK-RNA (752 bp) encodes NDK (*ndk* gene), which provides a substrate for RNA replication by converting cytidine diphosphate (CDP) into cytidine triphosphate (CTP). Therefore, Rep- and NDK-RNAs cooperatively replicate through the translation of their encoded enzymes. During replication, mutations occur, and occasional recombination generates a parasitic RNA that loses the gene region of either Rep- or NDK-RNA but replicates by exploiting replicase and NDK translated from other RNAs (Fig 1B). We expected that recombination or ligation between Rep- and NDK-RNAs facilitated by Qβ replicase [13–15] would result in the appearance of a linked RNA that harbors both *rep* and *ndk* genes (Fig 1B).

### Long-term replication experiments

To investigate whether linked RNAs with the two genes appear from unlinked Rep- and NDK-RNAs, we conducted long-term replication experiments. We repetitively performed (1) RNA replication by incubating the cooperative RNA replication system at 37˚C for 4 or 6 h in water-in-oil droplets, (2) diluted the droplet population typically 5-fold, and (3) induced fusion and division of droplets through vigorous mixing (Fig 1C). In step (3), a fresh translation system was supplied to the RNA population, and the RNA molecules were randomly redistributed among the droplets.

It should be noted that compartmentalization in this experimental procedure is transient because RNA molecules in different droplets can be mixed in every step of droplet fusion. More sustained compartmentalization could be achieved if compartments undergo growth instead of fusion, as implemented in previous theoretical studies [8–11]. Although these differences in compartmentalization may affect the evolutionary dynamics, both types of compartmentalization are conceivable in primitive compartments [20].

We measured Rep- and NDK-RNA concentrations after every replication step using quantitative PCR after reverse transcription (RT-qPCR) with sequence-specific primers. We also measured parasitic RNA concentrations in some rounds using native polyacrylamide gel electrophoresis (PAGE), as their sequences were unknown.

In a previous long-term replication experiment, we maintained RNA concentrations below a certain level (1 nM) by changing the dilution rate every round [12]. This was important because high RNA concentrations cause frequent RNA recombination, producing short parasitic RNA that replicates faster than Rep- and NDK-RNAs. The parasitic RNA propagates throughout the compartments by fusion-division of the compartments and disrupts the cooperation between Rep- and NDK-RNAs [12]. However, in the present study, we increased RNA concentrations to induce recombination and ligation, as these processes produce linked RNA. To achieve a higher RNA concentration while repressing the amplification of parasitic RNA as much as possible, we employed a new protocol to determine the dilution rate; the dilution rate was kept low (5-fold) until RNA concentration reached a particular level, around which parasitic RNAs often appear, and then increased to 100-fold for only a few rounds. We expected that this rapid dilution would circumvent the propagation of parasitic RNAs among the

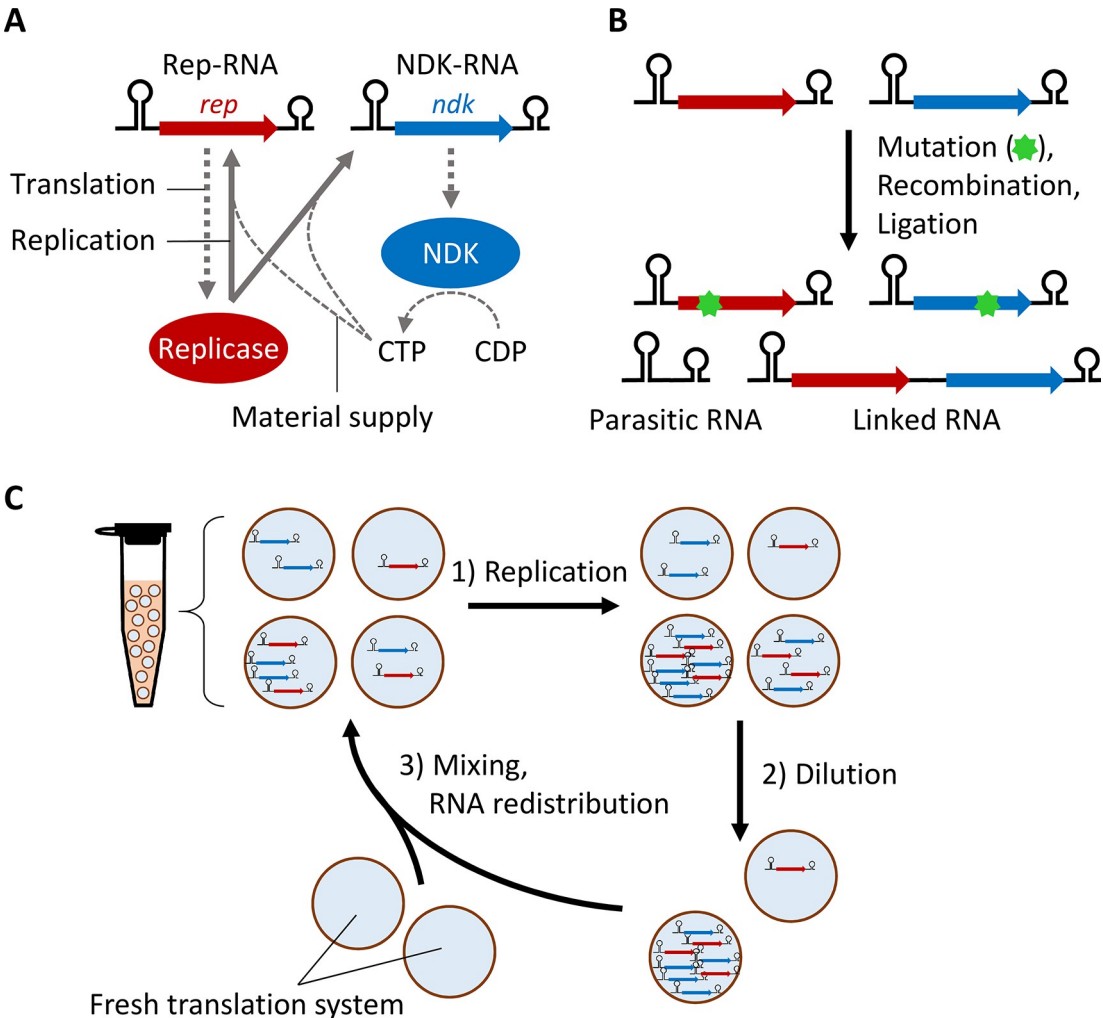

**Fig 1. Cooperative RNA replication system.** (**A**) Schematic representation of cooperative RNA replication. NDK is translated from NDK-RNA and converts CDP into CTP, a material for RNA replication, whereas replicase is translated from Rep-RNA and replicates both RNAs using the synthesized CTP. (**B**) During replication, mutant Rep- and NDK-RNAs and parasitic RNA that loses gene regions are generated by random mutation and recombination. Similar recombination or ligation could also generate linked RNA that harbors both *rep* and *ndk* genes. (**C**) Schematic representation of long-term replication experiments. 1) Cooperative RNA replication was performed at 37°C in water-in-oil droplets. 2) Droplets were diluted with new droplets containing a translation system. The dilution rate was typically 5-fold, but 100-fold dilution was used in some rounds. 3) Droplets were vigorously mixed to facilitate their random fusion and division, supplying the translation system to RNA, and RNA was randomly redistributed in the droplets.

compartments and allow both Rep- and NDK-RNAs to replicate continuously while periodically reaching high concentrations.

In the first long-term replication experiment, we initiated a serial dilution cycle with Rep- and NDK-RNAs obtained in our previous study (hereafter termed e1R0) [12]. In this experiment, we raised the dilution rate from 5- to 100-fold when the RNA concentration reached around 30 nM and successfully continued the replication of both Rep- and NDK-RNAs for 46 rounds (Fig 2A). Parasitic RNA concentrations were measured in four rounds with relatively high Rep- and NDK-RNA concentrations but barely detected (S1A Fig, the detection limit was approximately 10 nM). Next, we evaluated the presence of linked RNA in five rounds with relatively high Rep- and NDK-RNA concentrations by RT-PCR, using primers that could detect

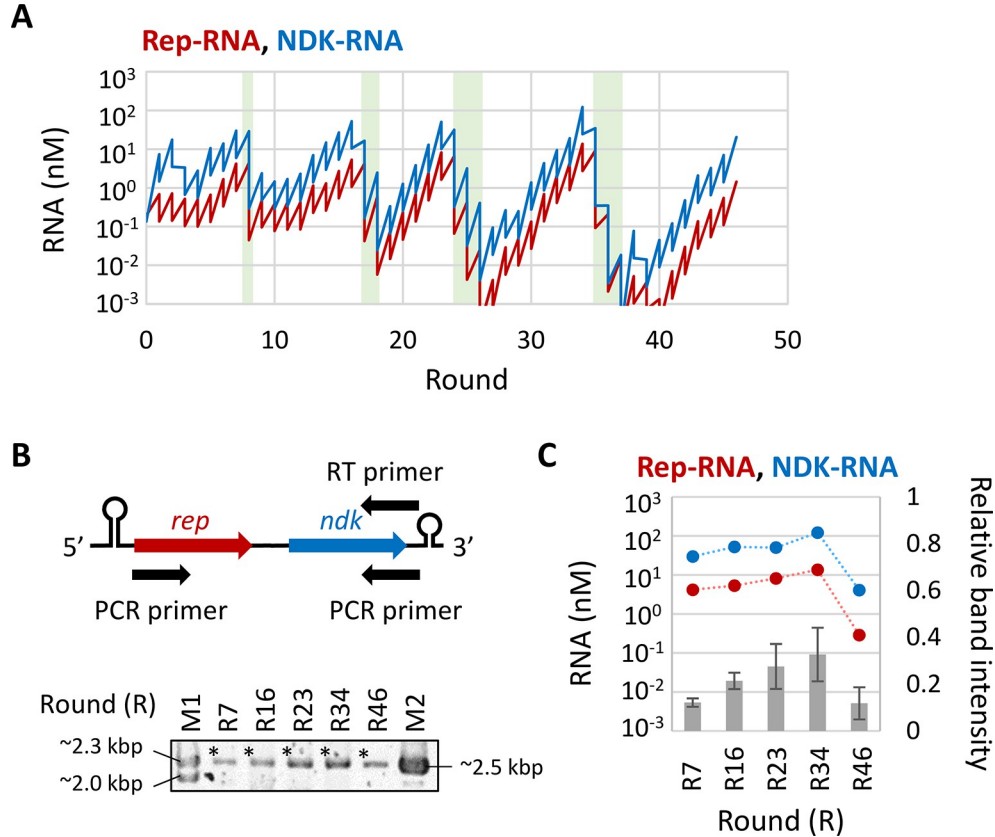

**Fig 2. The first long-term replication experiment.** (**A**) Changes in Rep-RNA (red) and NDK-RNA (blue) concentrations during the long-term replication experiment, measured by RT-qPCR. The experiment was initiated with Rep- and NDK-RNAs "e1R0" and conducted with high (100-fold) temporal dilution. The replication step was performed at 37°C for 6 h. Parasitic RNAs were visualized by native PAGE at rounds 8, 17, 24, and 34, but detected only at round 34 (S1A Fig). The parasitic RNA concentration at the round was 33 nM (not plotted). The light green areas highlight the rounds with high dilution. (**B**) RNA samples at the indicated rounds of the long-term replication experiment were subjected to RT-PCR, using primers that could detect 5′-rep-ndk-3′ (top), and PCR products were analyzed by agarose gel electrophoresis (bottom, a representative gel). M1 and M2 are size markers. Asterisks indicate analyzed bands. Repeated RT-PCR and negative control experiments were presented in S2 Fig. (**C**) Relative band intensities of the RT-PCR products to M2 (grey bars, right axis), in comparison with Rep- and NDK-RNA concentrations (red and blue plots, left axis). Error bars indicate standard errors (n = 4). Dotted lines are plotted for visibility.

RNA with *rep* and *ndk* genes linked in the order 5′-*rep-ndk*-3′ (Fig 2B). If the entire region of *rep* and *ndk* genes is retained in the linked RNA, the expected size of the PCR products is between 2136 and 2456 bp. Using agarose gel electrophoresis, we detected approximately 2.5 kbp products in all rounds (Fig 2B), with average band intensities roughly correlated with the base-10 logarithm of Rep- and NDK-RNA concentrations measured by RT-qPCR (correlation coefficients are 0.77 and 0.81, respectively) (Fig 2C). We also performed RT-PCR using primers for RNA with the two genes linked in the reverse order (5′-*ndk-rep*-3′) (S3A Fig), but barely detected approximately 2.5 kbp products (S3B and S3C Fig). These results suggested that longer RNAs that linked *rep* and *ndk* genes in a particular order appeared during the serial dilution cycle.

Encouraged by the first experiment, we performed a second long-term replication experiment with one of the mutant Rep- and NDK-RNAs obtained at round 46 of the first experiment. These mutant Rep- and NDK-RNAs accumulated six and one mutations, respectively

(S1 Table). The replication abilities of the mutant RNA pair (e2R0) were different from those of the original pair (e1R0); Rep- and NDK-RNA replications increased and decreased, respectively (S4 Fig).

In the second long-term replication experiment with the e2R0 pair, we changed the dilution rate from 5- to 100-fold when the RNA concentration reached around 100 nM and successfully continued the replication of both Rep- and NDK-RNAs for 79 rounds (Fig 3A). Parasitic RNA concentrations were measured in rounds 5–7, 20–22, 34–36, 54–56, and 60–62, in which Rep- and NDK-RNA concentrations were relatively high, and detected only in rounds 20–22 (Figs 3A and S1B). The dilution rate was not increased around rounds 35 and 62 when parasitic RNA was undetectable; however, Rep- and NDK-RNA concentrations decreased and then recovered. As a control, we also conducted a continuous replication experiment without changing the dilution rate when the RNA concentration reached 100 nM. In this case, parasitic RNA appeared and replicated excessively (S1C and S5 Figs), and then the concentrations of Rep- and NDK-RNAs decreased to undetectable levels, confirming the importance of changing the dilution rate.

Next, we examined the appearance of a linked RNA by RT-PCR every 1–3 rounds, using primers that could detect 5′-rep-ndk-3′ (Fig 3B). We detected approximately 2.3–2.5 kbp PCR products at 15 of 32 analyzed rounds; the products appeared in all three trials of RT-PCR at rounds 6, 19, 21, 22, 34, 35, 55, 61, 62, and 79 (S2 Fig). We also detected seemingly smaller (2.0–2.3 kbps) products at a fraction of rounds, although their appearance was stochastic (S2 Fig). The relative intensities of the bands, if detected, were as strong as those in the first long-term replication experiment (Fig 2B) throughout the rounds. The average band intensities were roughly correlated with the base-10 logarithm of Rep- and NDK-RNA concentrations (correlation coefficients are 0.71 and 0.82, respectively) (Fig 3C). The highest band intensity was observed at the final round (79), at which Rep- and NDK-RNA concentrations were not the highest, suggesting that the linked RNA may have been selected in the final round. The concentration of a potentially linked RNA in the population at round 79 was estimated to be less than 1 nM, which is less than 10% of either of the unlinked RNAs, by RT-qPCR using primers that could detect the linkage region of 5′-rep-ndk-3′. We also performed RT-PCR at rounds 6, 19, 34, 54, 62, and 79 using primers for 5′-ndk-rep-3′ (S3B and S3D Fig), and barely detected approximately 2.5 kbps products.

## Sequence analysis

Next, we obtained 47 clones of the potentially linked RNA products (5′-rep-ndk-3′) at round 79 of the second long-term replication experiment and analyzed their sequences. All analyzed clones harbored the entire region of both *rep* and *ndk* genes in the 5′-rep-ndk-3′ order, confirming that the obtained long RNAs were the linked products of Rep- and NDK-RNAs (hereafter referred to as RepNDK-RNA). We found two major ways by which Rep- and NDK-RNAs were linked. In 45% of the RepNDK-RNA clones, Rep- and NDK-RNAs were simply connected end-to-end without deletion or insertion. In 26% of the clones, a single C was inserted between the connected Rep- and NDK-RNAs. At the NDK-RNA site around the linkage, we also detected G3A and U12G in 83% and 87% of the RepNDK-RNA clones, respectively (Fig 4A, linkage region).

Next, to investigate the origin of RepNDK-RNA at round 79, we compared the sequences of the RepNDK-RNA clones with those of unlinked Rep- and NDK-RNAs. If RepNDK-RNA was continuously replicated during the long-term replication experiment, they could have accumulated mutations different from those found in unlinked Rep- and NDK-RNAs in the same round. Alternatively, if RepNDK-RNA appeared from Rep- and NDK-RNAs around round 79 or just as artificial products during the RT-PCR process, they should have a similar

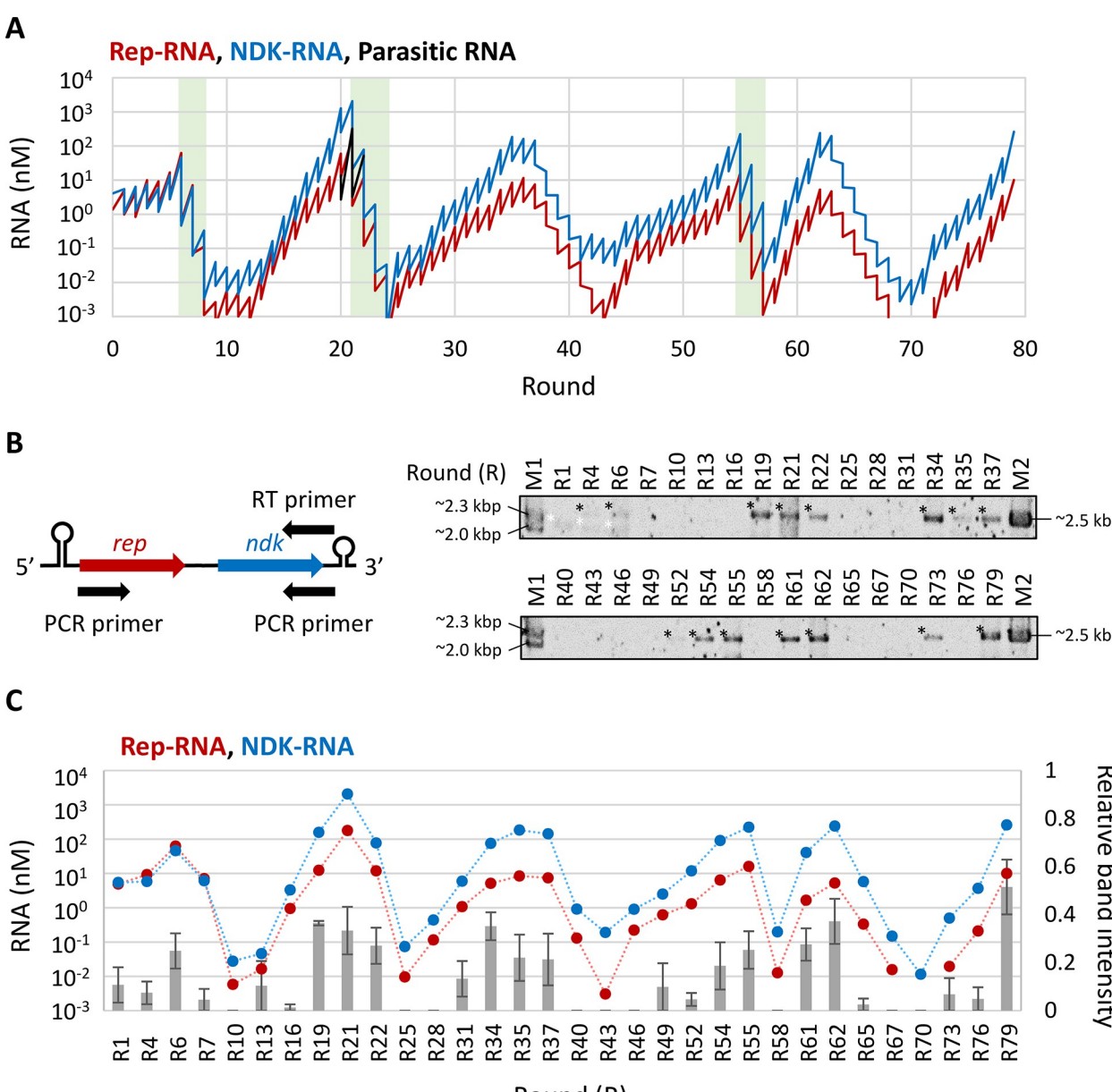

**Fig 3. The second long-term replication experiment.** (**A**) Changes in Rep-RNA (red), NDK-RNA (blue), and parasitic RNA (black) concentrations during the long-term replication experiment. The experiment was initiated with Rep- and NDK-RNAs "e2R0" and conducted with high (100-fold) temporal dilution. The replication step was performed at 37˚C for 4 h. RNA concentrations were measured by RT-qPCR (Rep- and NDK-RNAs) or based on native PAGE (parasitic RNAs, S1B Fig). Detection of parasitic RNAs was attempted at the rounds described in the main text. Rounds with high dilution are highlighted in light green. (**B**) Detection of putative linked RNAs that harbor both *rep* and *ndk* genes (5′-*rep*-*ndk*-3′) in the RNA samples in every 1–3 rounds of the long-term replication experiment. RT-PCR was performed using the indicated primers (left), and PCR products were analyzed by agarose gel electrophoresis (right, representative gels). M1 and M2 are size markers. Asterisks indicate analyzed bands; white ones indicate shorter sizes. Repeated RT-PCR and negative control experiments were presented in S2 Fig. (**C**) Relative band intensities of RT-PCR products to M2 (grey bars, right axis), in comparison with Rep- and NDK-RNA concentrations (red and blue plots, left axis). Error bars indicate standard errors (n = 3). Dotted lines are plotted for visibility.

set of mutations. To distinguish these possibilities, we obtained 47 clones of unlinked Rep- and NDK-RNAs at round 79 and analyzed their sequences. The Rep- and NDK-RNA clones contained 3–9 and 0–7 mutations, respectively (S1 Data).

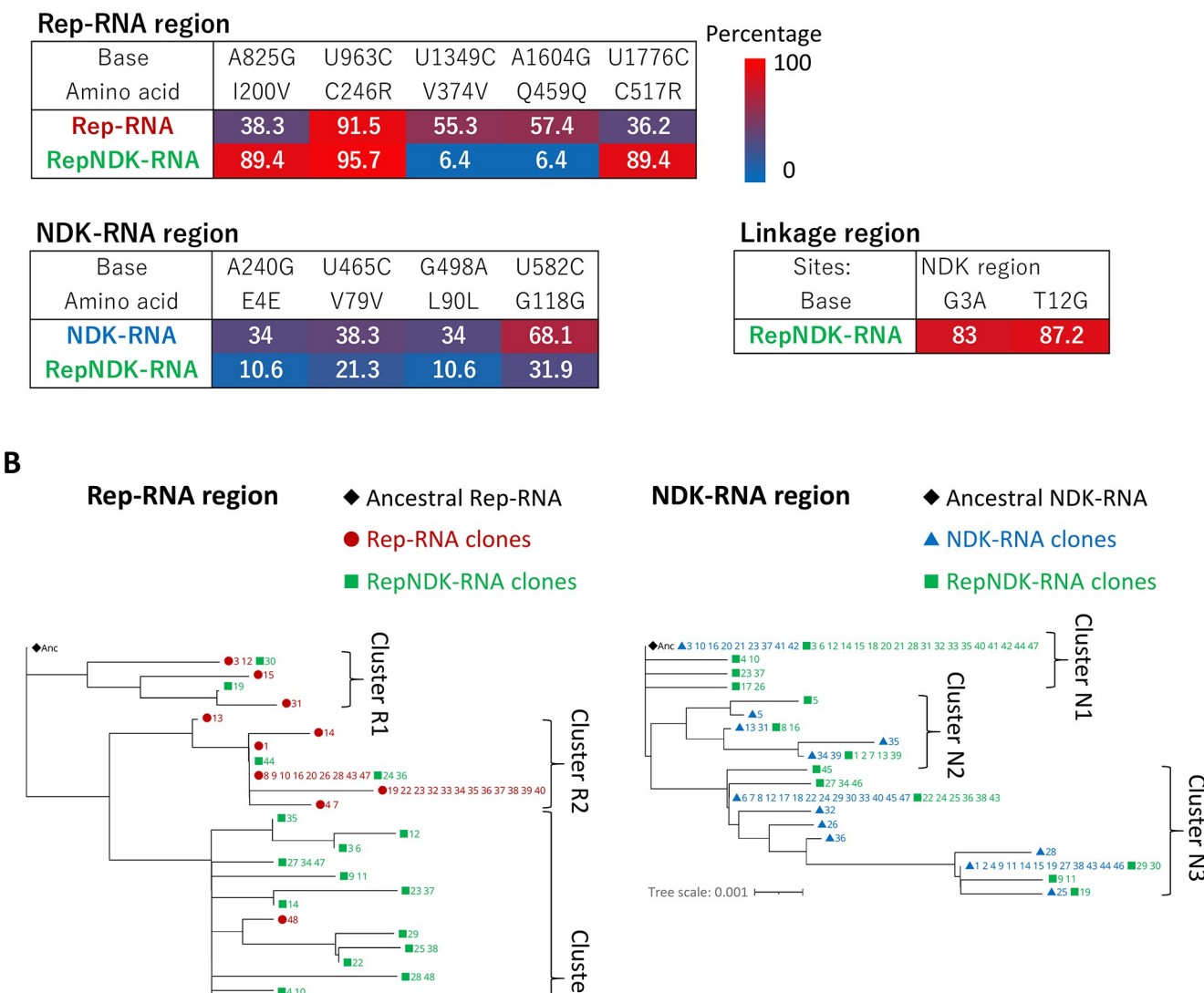

**Fig 4. Accumulated mutations in the linked and unlinked RNAs.** (**A**) Mutations detected in more than 30% of the 47 clones of any RNA species analyzed at round 79 are listed with their percentages. The sequence of the RepNDK-RNA clones was divided into Rep-RNA, NDK-RNA, and linkage regions for comparison with the Rep- and NDK-RNA clones. All detected mutations are listed in S1 Data. (**B**) Phylogenetic trees for each Rep- (left) and NDK- (right) RNA region were constructed from all the analyzed clones and mutations that were detected in at least two clones. Leaves representing different RNA species (Rep-, NDK-, or RepNDK-RNA) are marked with different colored symbols, followed by clone numbers. The ancestral Rep- and NDK-RNAs (Anc) were analyzed and displayed together.

Fig 4A lists all mutations detected in more than 30% of the clones of any of the three RNA species (unlinked Rep- and NDK-RNAs and linked RepNDK-RNAs). The three types of RNAs accumulated different mutations at their corresponding sites. For example, A825G and U1776C were found in 38% and 36% of the Rep-RNA clones, respectively, whereas both

mutations were detected in 89% of the RepNDK-RNA clones. In contrast, U1349C and A1604G were observed in 55% and 57% of the Rep-RNA clones, respectively, but detected in only 6% of the RepNDK-RNA clones. Similarly, A240G and G498A were detected in 34% of the NDK-RNA clones but in only 11% of the RepNDK-RNA clones. These different mutation patterns indicate that the majority of the RepNDK-RNAs analyzed at round 79 were not derived from Rep- and NDK-RNAs of the same round. These results also suggested that RepNDK-RNA was maintained in the population for a certain period, long enough to form its own lineage.

We further illustrated the evolutionary relationships between Rep- or NDK-RNA and RepNDK-RNA using phylogenetic trees for each of the Rep- and NDK-RNA regions (Fig 4B). The three RNA species are represented by different symbols. For the Rep-RNA region (Fig 4B, left), there were three clusters (Clusters R1–3). Cluster R1 consisted of four unlinked Rep-RNA clones (red circles) and two linked RepNDK-RNA clones (green squares). Cluster R2 mainly contained unlinked Rep-RNA clones; it consisted of 26 Rep-RNA clones and only three RepNDK-RNA clones. In contrast, Cluster R3 mainly contained linked RepNDK-RNA clones; it consisted of 17 Rep-RNA clones and 42 RepNDK-RNA clones. Furthermore, the unlinked Rep-RNA clones were located near the root of the branches in Cluster R3, compared to the linked RepNDK-RNA clones. These results suggested that RepNDK-RNA was forming distinct lineages from the unlinked RNAs in Cluster R3. For the NDK-RNA region (Fig 4B right), there were also three clusters, However, unlike the Rep-RNA region, the unlinked NDK-RNA and linked RepNDK-RNA clones were mixed well, indicating that the characteristic mutations of linked RNA were not present in the NDK-RNA region.

## Biochemical characterization of RepNDK-RNA

To understand the biochemical characteristics of the linked RepNDK-RNA at round 79, we selected a clone containing only the most common mutation set (A825G, T963C, U1776C, and -2042C in the Rep-RNA region and G3A and U12G in the NDK-RNA region), which was found in 26% of the 47 clones. For comparison, we also chose a pair of unlinked Rep- and NDK-RNAs containing only the most common mutation sets at round 79. The selected Rep-RNA contained U963C, U1349C, and A1604G, found in 55% of the Rep-RNA clones, whereas the selected NDK-RNA contained A240G, U465C, G498A, and U582C, found in 32% of the NDK-RNA clones. The replication of these Rep- and NDK-RNAs (e2R79) was similar to that of the ancestral Rep- and NDK-RNA pair (e2R0) with slightly improved NDK-RNA replication (S4 Fig).

RepNDK-RNA was expected to replicate by expressing both encoded replicase and NDK. To test this, we incubated the RepNDK-RNA clone in a reconstituted translation system containing fluorescent-labeled lysyl-tRNA. The newly synthesized proteins were analyzed by sodium dodecyl sulfate (SDS)-PAGE, followed by fluorescent imaging. We found that bands corresponding to both the replicase subunit and NDK were detected for the RepNDK-RNA (Fig 5A). The protein amounts estimated from the band intensities were approximately half of those of the Rep- and NDK-RNAs (Fig 5B). Next, we investigated whether the translation of the two proteins was coupled with RepNDK-RNA replication. During RNA replication, the replicase first synthesizes the minus (complementary) strand, which is then recognized by the replicase for plus-strand synthesis (Fig 5C). We incubated the RepNDK-RNA clone with a translation system in water-in-oil droplets at 37°C for 4 h and analyzed the synthesis of full-length plus and minus strands by RT-PCR. Agarose gel electrophoresis showed that the intensities of bands corresponding to both strands increased after incubation (Fig 5D), demonstrating the ability of the RepNDK-RNA to replicate without the help of unlinked RNAs. To

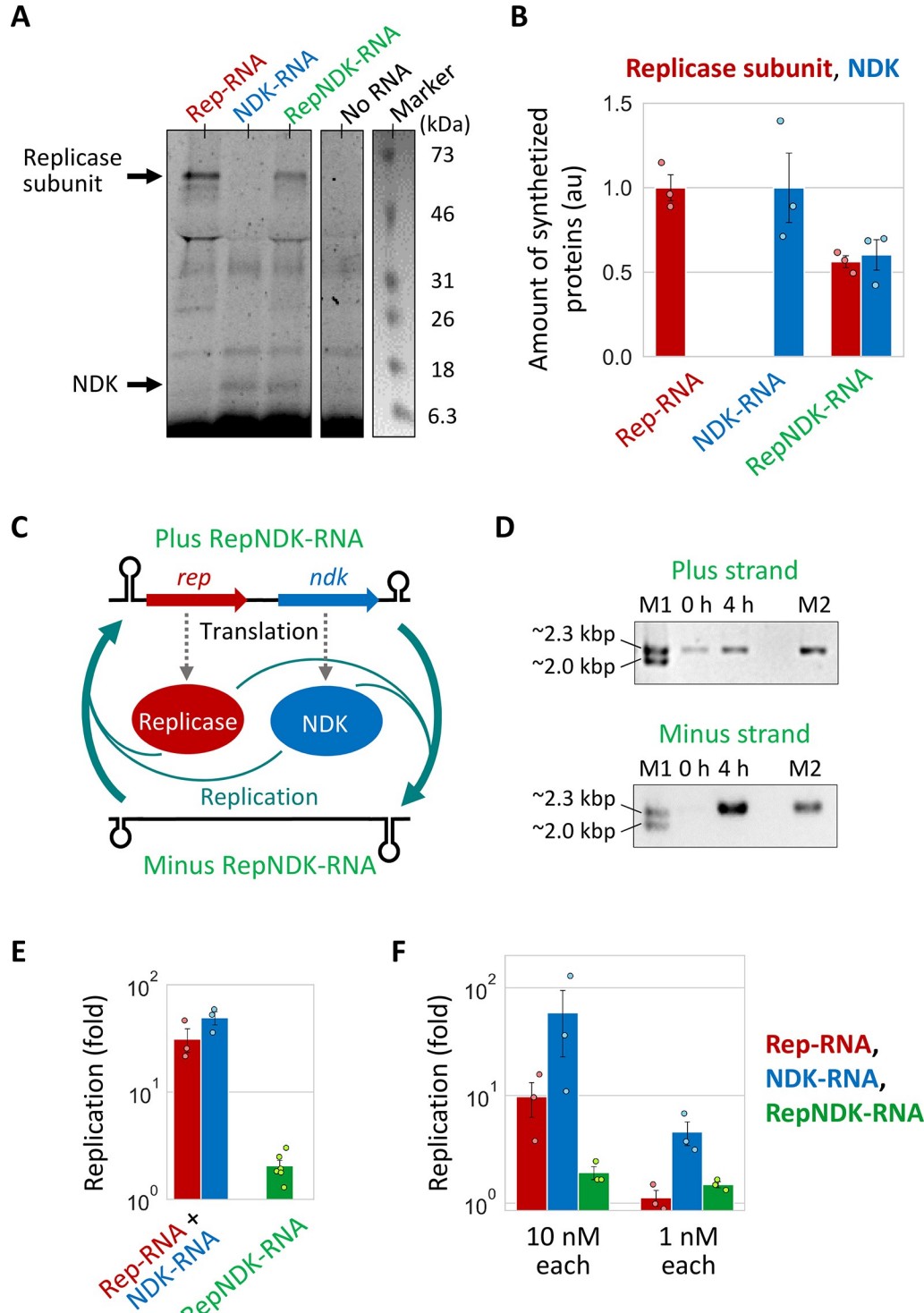

**Fig 5. Biochemical properties of the RNA clones at round 79.** (**A**) SDS-PAGE of the translation products after incubation of the Rep-, NDK-, and RepNDK-RNA clones (300 nM) at 37°C for 12 h in a modified translation system containing a fluorescently labeled lysine tRNA. Neither CTP nor CDP was present to prevent RNA replication. An analyzed fluorescent gel image is shown, in parallel to a trimmed white-light image of the same gel to visualize a pre-stained molecular weight marker. The expected bands of the replicase subunit (~64 kDa) and NDK (~15 kDa) are indicated by the black arrows. (**B**) The amounts of synthesized proteins, normalized to those of the Rep- or NDK-RNA clones (replicase subunit or NDK, respectively). Error bars indicate standard errors (n = 3). (**C**) The expected replication

scheme of RepNDK-RNA. The translation of the two encoded proteins induced RNA replication, completed with minus strand (complementary strand) synthesis from a plus strand (template) and following plus strand synthesis from the minus strand. (**D**) The RepNDK-RNA clone (10 nM) was incubated with the translation system in water-in-oil droplets at 37˚C for 4 h, and the replications of entire plus and minus strands were analyzed by RT-PCR and agarose gel electrophoresis. M1, size marker. M2, control PCR product of the RepNDK-RNA clone. (**E**) The replication amount of the RepNDK-RNA clone was measured by RT-qPCR in comparison with that of the Rep- and NDK-RNA pair (10 nM each). Error bars indicate standard errors (n = 3–6). (**F**) The mixture of the Rep-, NDK-, and RepNDK-RNA clones (10 nM each or 1 nM each) was incubated with the translation system in water-in-oil droplets at 37˚C for 4 h, and their replication amounts were measured by RT-qPCR.

quantitatively compare the replication of the linked RNA with that of the unlinked RNAs, we measured the replication of the RepNDK-RNA by RT-qPCR. The extent of RepNDK-RNA replication was less than 10% of Rep- and NDK-RNA replications (Fig 5E), which may be caused by the reduced translation activity.

Although the linked RepNDK-RNA clone replicated less efficiently than the pair of unlinked Rep- and NDK-RNAs, the linked RNA may have a relative selective advantage at low RNA concentrations because it can replicate on its own. We examined this possibility by incubating a set of Rep-, NDK-, and RepNDK-RNA clones mixed at low (1 nM each) or high (10 nM each) concentrations with a translation system in water-in-oil droplets at 37˚C for 4 h (Fig 5F). As expected, the replication of both Rep- and NDK-RNA clones decreased significantly at the 1 nM condition compared to the 10 nM condition (8.7- and 13-fold decreases, respectively). However, the RepNDK-RNA replication decreased only slightly (1.3-fold decrease) at the 1 nM RNA condition. Consequently, while the RepNDK-RNA clone replicated much less than the Rep- and NDK-RNA clones at the 10 nM condition, it replicated more than the Rep-RNA clone under the 1 nM condition. These results confirmed that RepNDK-RNA can be more competitive at lower RNA concentrations.

We also investigated the effect of the three dominant amino acid substitutions (I200V, C246R, C517R) in the replicase (Fig 4A) on RepNDK-RNA replication. We performed translation-uncoupled experiments in the following two steps. First, a modified Rep-RNA variant that contains each of the three non-synonymous mutations corresponding to the amino acid substitutions was incubated to induce translation of the replicase. In this reaction, the replication of the Rep-RNA was precluded because we removed the 3′ end (replicase recognition site) of the Rep-RNA and omitted CTP in the reaction mixture Second, we mixed the reaction mixture with the RepNDK-RNA and CTP to initiate RepNDK-RNA replication. We observed only negligible effects of any single amino acid substitutions on the ability of the replicase for RepNDK-RNA replication (S6 Fig). Thus, the non-synonymous mutations may have contributed to different factors such as the ability of RepNDK-RNA to be a template for replication.

## Discussion

The integration of distinct genetic information on a primitive chromosome has been considered a major evolutionary transition in the development of life [2–4]. In this study, we demonstrated that such a linkage emerged during the long-term replication of two cooperative RNAs encoding replication and metabolic enzymes in cell-like compartments. The linked RNAs that existed in the final population retained both genes, maintained their translation and replication abilities, and accumulated mutations different from those found in the unlinked RNAs (Figs 4 and 5), suggesting that the linked RNA continuously replicated in the RNA population for at least a certain period. These results provide experimental evidence that individual RNA replicators encoding different genes can assemble into a single RNA molecule through

Darwinian evolution. This process may explain the origin of primitive chromosomes during prebiotic evolution.

Although the accumulation of unique mutations in RepNDK-RNA indicated their maintenance in the population, at round 79, the linked RNA replicated less efficiently than the pair of Rep- and NDK-RNAs (Fig 5E), possibly because of its lower translational activity (Fig 5B), extended length, and reorganized RNA structures. How the linked RNA was maintained in the population while competing with the unlinked RNAs that replicate faster was not fully understood. One possible explanation could be that, unlike unlinked Rep- and NDK-RNAs, RepNDK-RNA can always replicate alone after redistribution in the droplet population, as suggested by theoretical studies [8,9,11]. This effect is expectedly enhanced when the stochastic mis-encapsulation of Rep- and NDK-RNAs becomes effective at low RNA concentrations [12] as supported by our experiment (Fig 5F). In addition, unlike co-replicating Rep- and NDK-RNAs, independently replicable RepNDK-RNA has another advantage of avoiding the competition between RNA replicators and balancing the translation of the two encoded proteins. Experiments using lower RNA concentrations for long-term replication could provide more selective advantages for RepNDK-RNA.

It should be noted that some theoretical studies assumed the putative RNA world, where RNA (ribozyme) acts as both a catalyst and a carrier of genetic information [10,11]. Thus, some theoretical predictions, such as the evolution of linked RNA to increase the copy number of ribozymes (i.e., dosage effect) [11] cannot be directly applied to our work in which RNA replicates using translated proteins. The translation process could also affect RNA replication in an unexpected way. For example, the competition between Qβ replicase and the ribosome to use the same RNA strand in opposite direction may inhibit efficient RNA replication [21], especially for long RNA molecules. Thus, the evolution of linkage may be more difficult than assumed in an RNA-only system.The generation of a linkage between RNA fragments is not unique to our cooperative RNA replication system. Diverse RNA-dependent RNA polymerases (RdRp), including Qβ replicase, are known to cause intermolecular RNA recombination during replication [22–24], which can link multiple RNA molecules. End-to-end ligation, as detected in the present study (with or without single nucleotide insertion), has also been observed for various RdRp such as through end-to-end template switching [14,25]. Thus, whether linked RNAs can evolve in other *in vitro* RNA replication systems using different RNA replicases [26] needs to be investigated. Furthermore, even before the emergence of proteins, such recombination and ligation of RNA molecules could have occurred spontaneously [16–18] or with the assistance of catalytic RNAs [27–31]. Therefore, it is reasonable to assume that the linkage of cooperative RNA molecules was prevalent during the early evolution of life.

We also developed a new method that allows the long-term replication of cooperative RNAs, even if the RNA concentrations increase tentatively and parasitic RNA emerges. Our previous study suggested that maintaining a relatively low RNA concentration was crucial for sustained replication of the cooperative system, and continuous RNA replication was achieved only by controlling the dilution rate at every replication round and maintaining the two cooperative RNAs below 1 nM [12]. However, the present study demonstrated that sustainable cooperative RNA replication does not require as strict regulation of RNA concentration as previously employed, and the cooperative RNAs could be more than 100 nM, at least occasionally. Thus, although tentative high dilution was still necessary, the molecular cooperation was more robust than previously thought. A higher RNA concentration also increases the chance of recombination or ligation of RNAs and may facilitate the evolution of a primitive chromosome.

The evolution of a linked RNA from multiple RNAs that encode different genes may also be useful for developing an artificial replicable RNA genome. To the best of our knowledge,

RepNDK-RNA is the first artificial RNA genome that replicates based on the translation of more than one encoded protein. The manual expansion of an RNA genome by introducing new genes is challenging, as it typically disrupts RNA structures and impairs its ability to undergo replication and translation. Our previous study connected two RNAs based on the predicted RNA structures of linked RNAs while maintaining their replication coupled with the translation of one encoded protein [32]. However, further introduction of genes would make genome expansion more difficult as the accuracy of RNA structural production decreases with increasing RNA length. Although further refinement is necessary, the evolutionary technique to integrate genes into a single RNA genome, as demonstrated here, could be utilized to create an optimized RNA genome encoding multiple proteins and contribute to the development of artificial cells that possess a genome replication system [33,34].

## Materials and methods

### Plasmids and RNAs

Two plasmids, each encoding Rep- or NDK-RNA "e1R0," were obtained in the previous study as the plasmids encoding Rep- or NDK-RNA "Evo" [12]. The other plasmids, each encoding one of the two Rep- and NDK-RNA clones ("e2R0" or "e2R79") or the RepNDK-RNA clone, were obtained in the present study through cloning, as described below, and site-specific mutagenesis. The plasmid encoding Rep-RNA clone e2R0 was further subjected to site-specific mutagenesis to introduce one of the three dominant amino acid substitutions (I200V, C246R, C517R). For the RepNDK-RNA clone, the entire cDNA length was PCR-amplified from the corresponding plasmid. The cDNA of the RepNDK-RNA clone and the plasmids for all other RNA clones except for ones with the amino-acid substitutions and a control RNA without substitutions were subjected to digestion with Sma I (Takara) and *in vitro* transcription with T7 RNA polymerase (Takara). The cDNAs of Rep-RNA variants containing specific amino-acid substitutions and the control Rep-RNA were subjected to PCR amplification for truncation of 40 bp at the 3′ end before *in vitro* transcription. All transcribed RNAs were purified using the RNeasy Mini Kit (QIAGEN).

### Preparation of the reconstituted translation system

The composition of the reconstituted translation system was as described previously [12] (based on the reconstituted *Escherichia coli* translation system [19]) except that the trigger factor was omitted because it was not essential for translation and contained a high level of NDK activity that could not be reduced by the following purification steps. All protein components of the translation system were purified by two successive affinity column chromatography in a stringent buffer to further reduce the remaining NDK activity derived from *Escherichia coli*. For all proteins except ribosomes, the re-purification procedure was the same as that used for removing tRNA from EF-Tu, described in our previous study [35]. Ribosomes were purified as described previously [19] and then washed with another stringent buffer (20 mM Hepes-KOH (pH 7.6), 6 mM magnesium acetate, 7 mM 2-mercaptoethanol, 1% Triton X-100, 1 mM dithiothreitol, and 0.33 M potassium chloride). Briefly, the ribosomes were diluted 20-fold with the stringent buffer and ultracentrifuged at 150,000 g for 2 h at 4°C. After removing the supernatant, the precipitate was rinsed with 70S buffer (20 mM Hepes-KOH (pH 7.6), 6 mM magnesium acetate, 7 mM 2-mercaptoethanol, and 0.03 M potassium chloride). After carefully removing the remaining buffer, the precipitate was dissolved in 70S buffer. Then, the solution was diluted again with the stringent buffer and collected by ultracentrifugation, as described above. To remove the residual stringent buffer, the final ribosome solution was diluted with

70S buffer 10-fold and then concentrated using Amicon Ultra (30 kDa cut, Merck) three times.

## Assay of remaining NDK activity

The remaining NDK activity in each protein component before and after the purification step was estimated as the activity to convert adenosine diphosphate (ADP) into adenosine triphosphate (ATP). First, each protein component (twice the concentration of that in the translation system) was incubated with 2.5 mM ADP and 1.25 mM CTP in a reaction buffer (100 mM Hepes-KOH (pH 7.6), 70 mM glutamic acid potassium salt, 0.375 mM spermidine, and 11 mM magnesium acetate) at 37°C for 2 h. Then, an aliquot of the reaction was used to determine the amount of synthesized ATP using ATP Assay Kit (Colorimetric/Fluorometric) (Abcam). The assay was performed at 37°C, and the fluorescence intensity was measured using Mx3005P Real-Time PCR System (Agilent Technologies) as the indicator of ATP synthesis (S7 Fig).

## Long-term replication experiment

The experiments were performed as described previously [12] with several modifications. The translation system (10 μL) containing certain concentrations of Rep- and NDK-RNAs "e1R0" (Fig 2A) or "e2R0" (Figs 3A and S5) were added to 1,000 μL of buffer-saturated oil and mixed vigorously with a homogenizer (POLYTRON PT 1300D, KINEMATICA) at 16,000 rpm for 1 min on ice to obtain water-in-oil droplets. The preparation of the saturated oil was described in the previous study [36]. The droplets were incubated at 37°C for 6 h (Fig 2A) or 4 h (Figs 3A and S5) to induce RNA replication through protein translation. After incubation, an aliquot of the droplets was diluted 5-fold or 100-fold with fresh buffer-saturated oil, as described in the main text and shown in the corresponding figures. The solution was mixed with 10 μL of the translation system and homogenized using the same method to obtain a new droplet population, followed by incubation at 37°C for 4 or 6 h for the next round of RNA replication. In Fig 2A, the original droplet population was diluted 100-fold using the same method as that before incubation to make the initial population (approximately 0.1 nM of Rep- and NDK-RNAs). Rep- and NDK-RNA concentrations were determined at every round by RT-qPCR with primers 1 and 2 (Rep-RNA) or 3 and 4 (NDK-RNA) (S2 Table). The measurement was performed after diluting the droplets 100-fold with 1 mM EDTA (pH 8.0) and using One Step TB Green PrimeScript PLUS RT-PCR Kit (Takara).

## Measurement of parasitic RNA concentration

In some rounds of the long-term replication experiments, the water phase containing RNA was collected from the droplets (100 μL) by centrifugation (22,000 g, 5 min). The recovered phase was mixed with diethyl ether (40 μL) and centrifuged (11,000 g, 1 min) to remove the diethyl ether phase. Then, RNA was purified with the RNeasy Mini Kit (QIAGEN) and subjected to 8% polyacrylamide gel electrophoresis in 1×TBE buffer. The fluorescence intensities of bands corresponding to parasitic RNAs were quantified using ImageJ (NIH) after staining with SYBR Green II (Takara). The concentrations of parasitic RNAs were determined from the intensities based on a dilution series of a standard parasitic RNA (s222 [37]).

## RT-PCR and sequence analysis

RNA samples were obtained from the long-term replication experiments as described above. Rep- and NDK-RNAs were reverse transcribed with primer 8, PCR-amplified with primers 7

and 8 (S2 Table), and separated using 0.8% agarose gel electrophoresis with E-Gel CloneWell (Thermo Fisher Scientific). The cDNA samples were cloned into a pUC19 vector (PCR-amplified with primers 11 and 12) using In-Fusion HD Cloning Kit (Takara). After transformation into *Escherichia coli*, the sequences of randomly selected plasmids were analyzed. Similarly, RepNDK-RNA was reverse transcribed with primer 10, PCR-amplified with primers 9 and 10, size-selected, and cloned using the same method. The 5′ and 3′ untranslated regions of the RepNDK-RNA clones could not be retrieved to distinguish them from Rep- and NDK-RNAs in the same RNA population. The sequence information of the linkage region was obtained only for the RepNDK-RNA clones. To detect putative elongated RNAs with *rep* and *ndk* genes linked in this order (Figs 2C and 3C) or the reverse order (S3 Fig), RT-PCR was performed with primers 9 and 10 or 13 and 14, respectively. PCR products were visualized by agarose gel electrophoresis and stained with SAFELOOK Green Nucleic Acid Stain (FUJIFILM), and band intensities were determined using ImageJ (NIH). All sequence data were available in S1 Table and S1 Data.

## Phylogenetic analysis

The sequences of the Rep-, NDK-, and RepNDK-RNA clones obtained at round 79 of the second long-term replication experiment (Fig 3A and S1 Data) were subjected to phylogenetic analysis. We analyzed only mutations that were detected at least in two clones. The RepNDK-RNA clones were divided into Rep-RNA and NDK-RNA regions and compared with the Rep- and NDK-RNA clones. Only the sequence regions of the RepNDK-RNA clones and their respective unlinked RNAs for which mutation information was commonly available were used for the analysis. Phylogenetic trees were created using the neighbor-joining method in MEGA11 [38] by assuming the same mutation rate for point mutations, single nucleotide deletions, and single nucleotide insertions. Phylogenetic trees were visualized using Interactive Tree Of Life (iTOL) [39].

## Translation-coupled RNA replication experiments

A pair of Rep- and NDK-RNA clones (10 or 1 nM each) and/or the RepNDK-RNA clone (10 or 1 nM) was incubated with the translation system in water-in-oil droplets at 37˚C for 4 h. The concentration of each RNA clone was determined by RT-qPCR as described above. Primers 5 and 6, 15 and 16, and 17 and 18 were used for the RepNDK-RNA clone, the Rep-RNA clone, and the NDK-RNA clone, respectively (S2 Table).

## Translation-uncoupled replication experiment

First, a truncated Rep-RNA (10 nM) containing a selected amino-acid substation was incubated at 37˚C for 12 h in the original translation system without CTP to induce the translation of the encoded replicase. Second, an aliquot of the reaction mixture was 5-fold diluted in the fresh mixture containing the translation system, 1.25 mM CTP, 30 μg/ml streptomycin, and 10 nM RepNDK-RNA clone, and subjected to further incubation. After 4 h incubation at 37˚C, the amount of RepNDK-RNA replication was measured by sequence-specific RT-qPCR as described above.

## Analysis of protein translation

300 nM of Rep-, NDK-, or RepNDK-RNA was incubated at 37˚C for 12 h in a translation system and FluoroTect GreenLys tRNA (Promega), with neither CTP nor CDP to preclude RNA replication. In this experiment, we used a translation system before excessive purification of

protein components, as described previously [40], to enhance translation efficiency and obtain measurable amounts of proteins. After translation, an aliquot was treated with 0.1 mg/mL RNase A (QIAGEN) at 37°C for 15 min to digest fluorescently labeled lysine tRNA. Then, the solution was incubated at 95°C for 4 min in SDS sample buffer (50 mM tris(hydroxymethyl) aminomethane hydrochloride (Tris-HCl, pH 7.4), 2% SDS, 0.86 M 2-mercaptoethanol, and 10% glycerol) and subjected to SDS-PAGE using a 10–20% gradient gel (Funakoshi, Japan). The synthesized proteins that incorporated fluorescently labeled lysine were visualized using FUSION-SL4 (Vilber-Lourmat), and band intensities were determined using ImageJ (NIH).

## Supporting information

**S1 Fig. Detection of parasitic RNAs during the long-term replication experiments.** (**A, B, C**) Native PAGE of RNA mixtures during the long-term replication experiments shown in Figs 2A (A), 3A (B), and S5 (C). P_a and P_b are controls of commonly appearing parasitic RNAs of known sizes. Asterisks indicate the bands whose intensities were quantified. The expected parasitic RNA bands and their sizes are shown on the right. ss, single-strand. ds, double-strand.
(TIF)

**S2 Fig. Detection of putative linked RNAs harboring *rep* and *ndk* genes in this order.** (**A**) Repeated RT-PCR using the primers that could detect 5′-*rep*-*ndk*-3′ for RNA samples in the first long-term replication experiments. The PCR products were analyzed by agarose gel electrophoresis. M1 and M2 are size markers. Asterisks indicate analyzed bands. (**B**) RT-PCR was performed three times in the same method for 0.1 nM RepNDK-RNA (P.C.) or a mixture of Rep- and NDK-RNAs (N.C., corresponding to 200 nM and 2000 nM in the long-term replication experiments, respectively). These RNAs were the representative clones obtained from the second long-term replication experiment. (**C**) RT-PCR was performed three times for the same P.C. and in the absence of RNA (N.C.). (**D**) Repeated RT-PCR for RNA samples in the second long-term replication experiments were performed and analyzed in the same method.
(TIF)

**S3 Fig. Detection of putative linked RNAs harboring *rep* and *ndk* genes in the reverse order.** (**A**) RNA samples in the long-term replication experiments were subjected to RT-PCR using the primers that could detect 5′-*ndk*-*rep*-3′. (**B**) The PCR products were analyzed by agarose gel electrophoresis for RNA samples of the first (e1) and second (e2) long-term replication experiment (Figs 2A and 3A). All analyzed gels are shown (n = 3). M1 and M2 are size markers. Asterisks indicate analyzed bands. (**C, D**) Relative band intensities of the RT-PCR products, derived from the first (C) and second (D) long-term replication experiments, to M2 (gray bars, right axis), in comparison with Rep- and NDK-RNA concentrations (red and blue plots, left axis). Error bars indicate standard errors (n = 3). Dotted lines are plotted for visibility.
(TIF)

**S4 Fig. Translation-coupled cooperative RNA replication experiment.** A pair of Rep- and NDK-RNAs (10 nM each) was incubated with a translation system in water-in-oil droplets at 37°C for 4 h, and their replication was measured by RT-qPCR. e1R0, e2R0 (e1R46), and e2R79 represent the ancestral RNA clones in the first long-term replication experiment (Fig 2A), the ancestral clones in the second long-term replication experiment (Fig 3A) (obtained at round 46 of the first experiment), and the clones obtained at round 79 of the second experiment, respectively. Error bars indicate standard errors (n = 3).
(TIF)

**S5 Fig. Long-term replication experiment without high temporal dilution.** Changes in Rep-RNA (red), NDK-RNA (blue), and parasitic RNA (black) concentrations during the long-term replication experiment. The experiment was initiated with Rep- and NDK-RNAs "e2R0" and conducted without high temporal dilution. The replication step was performed at 37˚C for 4 h. RNA concentrations were measured by RT-qPCR (Rep- and NDK-RNAs) or based on native PAGE (parasitic RNAs, S1C Fig).
(TIF)

**S6 Fig. Replicase activities on RepNDK-RNA replication analyzed by translation-uncoupled replication experiments.** The experiments were performed in two steps. (1) A Rep-RNA clone (10 nM) with the truncated 3′ end was incubated at 37˚C for 12 h in the absence of CTP to translate the replicase without RNA replication. Rep-RNA was constructed based on the ancestral Rep-RNA ("None") or those containing one of the three dominant non-synonymous mutations that correspond to "I200V", "C246R", and "C517R" amino acid substitutions. (2) The translated replicase was mixed with the RepNDK-RNA clone (10 nM) and CTP to induce replication at 37˚C for 4 h, while stopping further translation by the addition of streptomycin. The amount of RepNDK-RNA replication was measured by sequence-specific RT-qPCR. Error bars indicate standard errors (n = 3).
(TIF)

**S7 Fig. Contamination levels of NDK in the translation system.** (**A, B**) The contamination level of NDK in each protein component before (A) and after (B) re-purification was measured as the increase in fluorescence using the ATP assay (see Materials and Methods). The protein names with high levels of contamination are indicated. The thick black lines indicate background fluorescence. (**C**) The contamination level of NDK in the mixture of all protein components before or after purification. The data were compared with the activity of 16 nM and 1.6 nM NDK. (**D**) 10 nM Rep-RNA (e1R0) was incubated in the presence or absence of 10 nM NDK-RNA (e1R0) with the purified translation system in water-in-oil droplets at 37˚C for 4 h, and their replication was measured by RT-qPCR. Error bars indicate standard errors (n = 4).
(TIF)

**S1 Table. The list of mutations in the Rep- and NDK-RNA clones obtained at round 46 of the long-term replication experiment shown in Fig 2A.** The highlighted clones (Rep-RNA clone 6 and NDK-RNA clone 10) were used for further long-term replication experiments (Figs 3A and S5).
(TIF)

**S2 Table. The list of primers (from 5′ end to 3′end).**
(TIF)

**S1 Data. The list of mutations detected in all analyzed RNA clones.**
(XLSX)

**S2 Data. All uncropped gel images and source data for the individual panels.**
(XLSX)

## Acknowledgments

We are grateful to Ms. Nana Kuroda for technical support.

## Author Contributions

**Conceptualization:** Kensuke Ueda, Ryo Mizuuchi, Norikazu Ichihashi.

**Data curation:** Kensuke Ueda.

**Formal analysis:** Kensuke Ueda, Ryo Mizuuchi.

**Funding acquisition:** Ryo Mizuuchi, Norikazu Ichihashi.

**Investigation:** Kensuke Ueda, Ryo Mizuuchi, Norikazu Ichihashi.

**Methodology:** Kensuke Ueda, Ryo Mizuuchi, Norikazu Ichihashi.

**Project administration:** Ryo Mizuuchi, Norikazu Ichihashi.

**Supervision:** Ryo Mizuuchi, Norikazu Ichihashi.

**Validation:** Kensuke Ueda, Ryo Mizuuchi, Norikazu Ichihashi.

**Visualization:** Kensuke Ueda, Ryo Mizuuchi.

**Writing – original draft:** Kensuke Ueda, Ryo Mizuuchi.

**Writing – review & editing:** Kensuke Ueda, Ryo Mizuuchi, Norikazu Ichihashi.

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
