## [Decision Letter · Decision Letter 0]

11 Nov 2022

Dear Dr Mizuuchi,

Thank you very much for submitting your Research Article entitled 'Emergence of linkage between cooperative RNA replicators encoding replication and metabolic enzymes thorough experimental evolution' to PLOS Genetics.

The manuscript was fully evaluated at the editorial level and by independent peer reviewers. The reviewers appreciated the attention to an important problem, but raised some substantial concerns about the current manuscript. Based on the reviews, we will not be able to accept this version of the manuscript, but we would be willing to review a much-revised version. We cannot, of course, promise publication at that time.

Perhaps sequencing the PCR products from Figures 2C, 3C, S2C, and S2E mentioned by reviewer #2 would remove doubts about their identities.

If you decide to revise the manuscript for further consideration at PLOS Genetics, please aim to resubmit within the next 60 days, unless it will take extra time to address the concerns of the reviewers, in which case we would appreciate an expected resubmission date by email to plosgenetics@plos.org.

We are sorry that we cannot be more positive about your manuscript at this stage. Please do not hesitate to contact us if you have any concerns or questions.

Yours sincerely,

Juergen Brosius

Academic Editor

PLOS Genetics

Kirsten Bomblies

Section Editor

PLOS Genetics

Reviewer's Responses to Questions

**Comments to the Authors:**

Reviewer #1: This is a very stimulating paper. It provides confirmation of theory, but it also enriches the picture with valuable results. It deserves to be published. Some important comments for a minor revision:

1. Emphasize that this is transient compartmentation. Fusion-fission cycles do not fit the model of a growing protocell population, as analyzed theoretically. This is important because the latter provides the most stringent selection for cooperation. The system used in this paper is transient, not only dynamically but also, presumably, in the early history of replicators. This is fully okay, but needs to be explained somewhat.

2. Chromosomes not only reduce the assortment load, but also automatically eliminate competition between the LINKED replicators. And we do see that the two unlinked genes are competing. Furthermore, linkage presumably ensures a balanced production of translation products., which is likely to advantageous. Discuss!

3. Going through proteins is different from the assumptions of the theory of being at an RNA world stage. In the latter case theory found that chromosomes with multigene families appear, but this is due to a direct dosage effect when RNAs act as enzymes -- not only as information carriers, as in the present paper.

3. This leads me to the big unknown: the translation products. The expressed proteins evolve because their genes evolve, but here we see only the analysis of the latter. Granted one cannot do everything in a paper, but please tocuh upon this important point in the Discussion. I hope you will do this missing work sometime later. Note that the RNA for the replicase component can affect its own replicative phenotype as template as well as through the catalytic phenotype of its protein product.So far we see a compound effect.

4. Finally, it is appropriate to call attention to the fact that these replcators use a highly evolved, complex translation system provided by the experimenterm which can introduce unwanted side effects.

But on the whole this is an important expperimental result.

Reviewer #2: In this article, Ueda et al. explore a very interesting idea concerning the emergence of primordial genomes. As the authors mention in the introduction, emergence of genomes would have conferred unique advantages and disadvantages to primordial replicators. They discuss these ideas briefly in the introduction and present the outline and objectives of the study in a clear and concise manner. The authors claim to demonstrate evolution of a ‘linked’ RNA genome, comprising of two (previously established) cooperative RNA replicators. The premise and relevance of the idea explored is relevant for audience interested in genetic and genomics research, particularly in the context of origin of life and evolution. However, I have multiple concerns and questions regarding the experimental methodology and data, as well as their description. Addressing these aspects is, in my opinion, necessary to improve the rigour of the manuscript, before it can be considered for acceptance in its current form.

1. The random emergence and disappearance of linked RNAs implies that all of the linked products do not bring about any fitness advantage (which seems to me the main selling point of the manuscript). This observation is also reflected in the lower replication rate and translation efficiency of the linked RNAs. As the authors suggest, the linked RNAs will likely be more competitive at higher RNA dilutions. The manuscript would greatly benefit if some simple competition experiment between linked and unlinked RNA species were performed at increasing levels of dilution.

2. For the relative RT-PCR band intensities of the linked products in Figures 2C, 3C, S2C, and S2E, at least three replicates should be performed if the authors want to display bars. Otherwise, I fear that the statistics are insufficient for this type of semiquantitative analysis and can, at most, only be used for a binary yes/no detection. In light of the seemingly random nature of the emergence of the RT-PCR products, repeated negative controls might also be necessary to rule out sample cross-contaminations or RT-artefacts such as template switching.

3. Given the huge population of sequences in each reaction, it seems questionable to me to draw phylogenetic conclusions about the origin/maintenance of the linked RNAs from the mutation patterns of only a handful of clones (6 to 16). In my opinion, the authors should either confirm their findings using high-throughput sequencing experiments (as they have already done in the past) or at least significantly increase the number of conventional clones they analyse.

4. More quantitative details on the RNA concentrations is sorely needed throughout the manuscript. E.g. In lines 144, 172 and 175., the amount of RNA is described in very vague terms only, which affects the scientific rigour of the manuscript.

5. What are the effects of the I200V, C246R, and C517R amino acid substitutions on the activity of Rep? Are they associated with a fitness advantage with respect to the replication of the linked genomes?

Apart from these, I have some minor comments:

- Figure S5. To me it seems unclear to which extend the arbitrary fluorescence increase is associated with residual NDK activity as there are no reference points (baseline / positive control). I.e. is the residual activity observed after re-purification still significant in particular if all proteins are combined again? Is there still detectable replication of Rep-RNA in this background without NDK-RNA?

- sentence in line 295-297. Isn’t this counter to experimental findings, considering the emphasis authors place on the dilution step? Please clarify.

**Have all data underlying the figures and results presented in the manuscript been provided?**

Reviewer #1: Yes

Reviewer #2: **No: **No uncropped gels were provided (Figs 2B, 3B, 5A, 5D, plus associated supplementary figures)

No source data for the inidividual plots (Figa 2A, 2C 3A, 3C, 5B, 5E, plus associated supplementary figures)

PLOS authors have the option to publish the peer review history of their article (what does this mean?). If published, this will include your full peer review and any attached files.

Reviewer #1: No

Reviewer #2: No

---

## [Decision Letter · Decision Letter 1]

18 Jul 2023

Dear Dr Mizuuchi,

We are pleased to inform you that your manuscript entitled "Emergence of linkage between cooperative RNA replicators encoding replication and metabolic enzymes through experimental evolution" has been editorially accepted for publication in PLOS Genetics. Congratulations!

Yours sincerely,

Juergen Brosius

Academic Editor

PLOS Genetics

Kirsten Bomblies

Section Editor

PLOS Genetics

Comments from the reviewers (if applicable):

Reviewer's Responses to Questions

**Comments to the Authors:**

Reviewer #1: Nice paper.

Reviewer #2: The authors have addressed all my concerns comprehensively and I support the publication of the revised manuscript.

**Have all data underlying the figures and results presented in the manuscript been provided?**

Reviewer #1: Yes

Reviewer #2: Yes

PLOS authors have the option to publish the peer review history of their article (what does this mean?). If published, this will include your full peer review and any attached files.

Reviewer #1: **Yes: **Eors Szathmary

Reviewer #2: No

**Data Deposition**

http://datadryad.org/submit?journalID=pgenetics&manu=PGENETICS-D-22-01166R1

**Press Queries**

---

## [Editor Report · Acceptance letter]

1 Aug 2023

PGENETICS-D-22-01166R1 

Emergence of linkage between cooperative RNA replicators encoding replication and metabolic enzymes through experimental evolution 

Dear Dr Mizuuchi, 

We are pleased to inform you that your manuscript entitled "Emergence of linkage between cooperative RNA replicators encoding replication and metabolic enzymes through experimental evolution" has been formally accepted for publication in PLOS Genetics! Your manuscript is now with our production department and you will be notified of the publication date in due course.

With kind regards,

Anita Estes

PLOS Genetics

On behalf of:
